# Food Consumption Inequalities in Primary Care in a Large Metropolis

**DOI:** 10.3390/ijerph21070935

**Published:** 2024-07-18

**Authors:** Mariana Souza Lopes, Priscila Lenita Candida dos Santos, Aline Cristine Souza Lopes

**Affiliations:** 1Nutrition Department, Health Sciences Center, Campus I, Universidade Federal da Paraíba, João Pessoa 58051-900, Brazil; marianalopes.ufpb@gmail.com; 2Nutrition Department, School of Nursing, Health Campus, Universidade Federal de Minas Gerais, Belo Horizonte 31270-901, Brazil; priscila.lenita@yahoo.com.br

**Keywords:** primary care, NOVA classification, food consumption, health inequality, vulnerability in health

## Abstract

The aim of this study was to examine the association between health vulnerability and food consumption according to the NOVA classification within primary care in a major Brazilian city. A cross-sectional study was conducted among adults over 20 years old. These participants were part of a representative sample from the Health Academy Program (PAS) in Belo Horizonte, Brazil. We evaluated socio-demographic variables, self-reported illnesses, perceived health and quality of life, and the length of participation in PAS. Health vulnerability was gauged through the Health Vulnerability Index (HVI), which is calculated for each census sector and classified as low, medium, and high/very high. On the other hand, food consumption was determined by evaluating the average consumption described in a 24 h diet recall (24HR) and categorizing it under the NOVA classification: culinary preparations, processed foods, and ultra-processed foods (UPFs). The average calorie intake was 1429.7 kcal, primarily from culinary preparations (61.6%) and UPFs (27.4%). After adjustments, individuals residing in high/very high-HVI areas consumed more culinary preparations (β = 2.7; 95%CI: 4.7; 0.7) and fewer UPFs (β = −2.7; 95%CI: −4.7; −0.7) compared to those from low-vulnerability areas. PAS participants residing in more vulnerable areas reported healthier dietary habits, consuming more homecooked meals and fewer UPFs. These findings underscore the importance of concentrating efforts on promoting and preserving healthy eating habits and emphasizing the value of home cooking in the most vulnerable regions.

## 1. Introduction

Recent years have seen significant changes in dietary habits, both in Brazil and globally. Notably, there has been a decrease in the consumption of fresh food and minimally processed meals, in contrast to the rising intake of ultra-processed foods (UPFs) [1,2,3].

Changes in diet and new eating patterns seem to be correlated with individuals’ social and economic vulnerabilities [2]. Research suggests that diet quality often deteriorates as income and other vulnerability indicators, such as education level, worsen. For instance, the Australian Health Survey (2011–2013) demonstrated that the consumption of UPFs was the highest among adults with lower education levels and within the poorest household income brackets [4]. A similar pattern has been noticed in the US Behavioral Risk Factor Surveillance System’s cross-sectional analysis, indicating a lower likelihood of individuals achieving the recommended daily intake of fruits and vegetables in states with significant income inequalities [5]. In a multinational study called the “Latin American Health and Nutrition Study (ELANS)”, which included participants from Argentina, Brazil, Chile, Colombia, Costa Rica, Ecuador, Peru, and Venezuela, it was found that individuals with a lower socioeconomic status consumed fewer fruits, vegetables, whole grains, fiber, fish, and seafood compared to their wealthier counterparts [6].

In Brazil, socioeconomic factors such as income and education significantly influence dietary habits, according to national surveys [1,7]. These factors are also associated with ethnic and racial disparities. For instance, individuals of brown or black race/skin color, classified according to the Brazilian institute of geography and statistics in their demographic censuses, who typically possess less income and education, are less likely to consumed diverse, healthy foods in rural regions of Brazil [8]. Furthermore, individuals residing in economically disadvantaged areas often face challenges accessing high-quality, nutritious food due to lower variety and quality, higher prices, and limited choices, particularly in the consumption of fruits and vegetables [9,10].

Socio-economic disparities in the area can affect the community and consumer food environment, and thus the quality, availability and price of food available in food stores for economically disadvantaged groups, particularly in Primary Health Care (PHC) equipment, i.e., the community’s first line of health defense. For example, fruits and vegetables available in disadvantaged areas when compared to affluent areas may have worse sensory qualities such as appearance, aroma, and consistency, in addition to a higher price. It is crucial to investigate these disparities in PHC users’ food consumption in order to identify highly vulnerable groups and plan effective nutritional interventions that promote healthy diets. However, this topic seldom experiences in-depth exploration, especially among Health Promotion Program participants who routinely participate in healthy eating activities. Consequently, this article seeks to investigate the relationship between health vulnerability and food consumption—following the NOVA classification—among PHC Health Promotion Program enrollees in a major Brazilian metropolis.

## 2. Materials and Methods

### 2.1. Study Design and Location

A cross-sectional study was conducted using baseline data from a randomized controlled community trial (RCCT) at the Health Academy Program (PAS) in Belo Horizonte, Minas Gerais, Brazil. The trial is registered with the Brazilian clinical trials under the URL RBR-8t7ssv at www.ensaiosclinicos.gov.br/rg/RBR-9h7ckx/ (accessed 10 June 2024).

At present, 81 PAS units are scattered across Belo Horizonte, capable of serving up to 20,000 individuals [11].

The PAS has been part of Brazil’s PHC since 2011, established with the goal of fostering equity in health promotion and care initiatives. To achieve this objective, it comprises units equipped with infrastructure and skilled professionals dedicated to promoting healthy lifestyles, with a particular emphasis on encouraging physical activity and promoting healthy eating. This is achieved through integration with the healthcare system [12].

### 2.2. Study Sample

The study employed simple cluster sampling, appropriately stratified across the nine administrative regions of the municipality. Eligible PAS units for inclusion in the study were required to operate in the morning and be situated in areas with medium-to-very high health vulnerability index (HVI) features predominant in the municipality. Units that had participated in food and nutrition research within the two years leading up to the study were excluded. Uniting these criteria, 42 PAS units were eligible. We selectively drew 18 units, with each region contributing two units matched by their HVI. This yielded a final sample that accurately represented PAS units with a medium, high, or very high HVI. The confidence level was at a significant 95%, and the margin of error was a low 1.4% [13].

We invited all participants aged 20 years or older who had attended health services in the last month (per the attendance list) to partake in the study. However, we excluded pregnant women and individuals with cognitive deficits that made it impossible to respond to the questionnaire.

In this study, also we excluded participants lacking HVI data (n = 320; 9.4%) and those with an extreme energy consumption (<500 kcal/day or >7000 kcal/day; n = 38; 1.1%) based on criteria proposed by Willett (2013) [14]. Thus, we analyzed data from 3056 individuals, accounting for 89.5% of the sample who responded to the survey.

### 2.3. Data Collection

From March 2013 to June 2014, trained interviewers conducted in-person interviews. The questionnaire, based on national studies and the research group’s prior experience, incorporated socio-demographic factors, health conditions, dietary habits, and anthropometric data (weight and height) [13].

### 2.4. Variables Studied

#### 2.4.1. Explanatory Variable of Main Interest: Health Vulnerability Index (HVI) of the Census Sector in Which the User Resides

The Health Department of Belo Horizonte provided the HVI data, which pertains to the research conducted in 2012 [15]. From the complete address, the participants’ census tracts of residence were identified, and the respective HVI values and classification were assigned.

The HVI is derived from eight variables grouped into two categories: sanitary and socioeconomic. The sanitary category includes variables such as inadequate water supply, sanitation, and garbage collection. The socioeconomic category considers the number of residents per house, the population’s literacy rate, the average income, the percentage of people earning up to half the minimum wage, and the percentage of indigenous and Black populations. The index ranges from 0 to 1, with higher values indicating greater health vulnerability within the area. This index is organized into four classifications based on average and standard deviation (SD) values: low (values below the average HVI), medium (average +/− 0.5 SD), high (values above the average HVI + 1 SD), and very high (values exceeding the high HVI). For this study, the “high” and “very high” classifications were combined due to a minimal number of census areas classified as having a “very high” HVI.

#### 2.4.2. Outcome Variable: Food Consumption According to the NOVA Classification

The average food consumption data were derived from two non-consecutive sources: a 24 h diet recall (24HR) used in conjunction with homemade food measurement kits. The homemade measuring kit consisted of different cutlery, glasses and plates of different sizes. Its application was carried out to better approximate the amounts ingested by participating individuals. In cases where only one 24HR was available for analysis (206 cases or 6.0%), this single data point was used as the participant’s average consumption [13].

The food quantities gathered using the 24HR method were converted into grams using table of measurements referred to for foods consumed in Brazil and Tables of nutritional composition of foods consumed in Brazil proposed by the Brazilian Family Budget Survey [13]. For foods not mentioned in these tables, we either used the information provided on their labels or directly weighed them with the assistance of a team trained at the University’s Dietetic Techniques Laboratory. Following this, we processed the food quantity data using Brasil Nutri, a software commonly used in Brazilian national surveys [13].

The consumed foods reported by participants in the 24HR were categorized based on the NOVA classification as proposed by MONTEIRO et al. [16], which organizes food items by their level and degree of industrial processing: unprocessed (e.g., offal, eggs, milk, seeds, fruits, leaves, stems, roots), minimally processed (include fresh, squeezed, chilled, frozen, or dried fruits and leafy and root vegetables; grains such as brown, parboiled or white rice, corn cob or kernel, wheat berry or grain; legumes such as beans of all types, lentils, etc.), processed culinary ingredients (e.g., such as oil, salt, sugar, herbs, spices), processed (include cheese, canned vegetables, salted nuts, fruits in syrup, and dried or canned fish), and UPFs [17]. UPFs are industrial formulations composed mostly or entirely of substances extracted from food, derived from food constituents, or synthesized in the laboratory based on organic materials (e.g., dyes, flavorings, and flavor enhancers) [16].

In the present study, fresh foods, minimally processed foods, and processed culinary ingredients were collectively labeled as culinary preparations [17].

In this evaluation, each food group’s contribution to total dietary energy was calculated and expressed as a percentage with the following formula: (calories from the food group × 100)/total calories (total energy intake).

#### 2.4.3. Covariates

The analyzed covariates were composed of socio-demographic, health, and nutritional status data. The socio-demographic data included gender (female, male); age; marital status (categories: married, separated/divorced, single, widowed); occupation (options: housekeeper, retired/pensioner, unemployed, employed); years of education; and per capita family income in reais, derived from the total family income divided by the household size.

We analyzed health issues, including reported illnesses (diabetes mellitus—DM and arterial hypertension—AH; yes/no), health perception (very bad/bad, fair, good/very good), and quality of life (very bad/bad, fair, good/very good). Additionally, we examined the duration of each individual’s participation in the PAS, calculated by subtracting the participant’s entry date into the health service from the interview date (in months).

The body mass index (BMI), calculated as weight (kg) divided by height (m) squared, was used to evaluate the nutritional status. Based on the BMI, individuals were categorized as underweight (≤18.5 kg/m^2^), normal weight (18.5–24.9 kg/m^2^), overweight (25–29.9 kg/m^2^), or obese (≥30 kg/m^2^) [12].

### 2.5. Statistical Analyses

The data were analyzed using data analysis and statistical software. Socio-demographic and health characteristics aligned with the HVI were outlined using linear trend statistical tests for categorical variables and the Kruskal–Wallis test, supplemented by Scheffe post hoc, for continuous variables. Food consumption in accordance with HVI was described using the analysis of variance (ANOVA) statistical test, modified with Scheffe correction.

We used multivariate linear regression to confirm the correlation between the HVI and the outcomes using five unique models: Model 0—Unadjusted; Model 1—Adjusted for sex, age, and education; Model 2—Adjusted with Model 1 variables plus time spent in PAS; Model 3—Adjusted with Model 2 variables, as well as health perception; Model 4—Adjusted with Model 3 variables and BMI. The models were individually developed for each outcome.

## 3. Results

Of the 3056 participants, 10.2% resided in regions with a low HVI, 56.7% in areas with a medium HVI, and 33.0% in high/very high-HVI areas. The majority of the participants were females (87.8%), with an average age of 58 years, eight years of education, a per capita family income of BRL 678.00, and had been in PAS for about 16.7 months (Table 1).

Upon analyzing socio-demographic characteristics in relation to the HVI, it was observed that individuals residing in high/very high-HVI areas were generally younger (56 years old versus 61 and 58 years old, *p* < 0.001) in comparison to those in the medium and low HVI. These individuals, residing in high/very high-HVI areas, had less education (an average of 5 years of study compared to 11 and 8 years, *p* < 0.001) and earned less income (BRL 600.00 against BRL 1000.00 and BRL 700.00, *p* < 0.001) compared to their counterparts. They also had a longer period of involvement in the PAS (with a median duration of 17.9 months in contrast to 11.1 and 16.9 months, *p* < 0.001) compared to individuals residing in areas of low and medium vulnerability, respectively (Table 1).

In terms of health conditions and nutritional status, 16.9% of respondents reported having diabetes (DM) and 53.2% reported arterial hypertension (HA), with no significant differences based on the HVI. A majority of participants (74.0%) perceived their health and quality of life as very good or good. However, this perception was less common among individuals living in areas with medium and high/very high-HVI compared to those in low-HVI areas (health perception: 74.1% and 71.5% vs. 82.7%, respectively; *p* < 0.001; quality of life: 81.5% and 75.6% vs. 82.4%, respectively; *p* < 0.001).

Regarding nutritional status, 40.2% of participants were overweight, with a higher prevalence among individuals residing in areas with a medium HVI compared to those in low-HVI areas (40.6% vs. 41.2%; *p* = 0.022) (Table 2).

The average daily energy intake was 1429.7 kcal, with the majority derived from culinary preparations (61.6%), followed by UPFs (27.4%) and processed foods (10.9%). When comparing individuals residing in areas with a high/very high HVI to those in low-HVI areas, it was observed that the former had a lower average energy intake (1382.9 kcal vs. 1497.9 kcal; *p* = 0.001), as well as lower consumption of processed foods (140.9 kcal vs. 159.9 kcal; *p* = 0.004) and UPFs (374.6 kcal vs. 450.0 kcal; *p* < 0.001) (Table 3).

After adjusting the multivariate models, we found a higher consumption of culinary preparations (β = 2.7; 95%CI: 0.7; 4.7; *p* = 0.007) and a lower consumption of UPFs (β = −2.7; 95%CI: −4.7; −0.7; *p* = 0.007) in individuals residing in areas with a very high/high HVI compared to those in areas with a low HVI (Table 4).

## 4. Discussion

The study demonstrated differences in food consumption amongst Brazilian PHC Health Promotion Program participants. Those residing in areas of higher health vulnerability reported increased consumption of culinary preparations and a decreased intake of UPFs. It was also observed that an individual’s socio-demographic and health characteristics were associated with health vulnerability.

Prior evidence shows that an area’s vulnerability can influence socioeconomic and health issues, including increased exposure to risky behaviors and chronic diseases, as well as limited access to and consumption of healthy foods [8,18,19].

The Health Inequalities data repository reveals disparities in health conditions and experiences based on differences in age, sex, economic status, education level, and place of residence, among others worldwide [20].

In Brazil, data from the National Health Survey [21] and the Surveillance Survey of Risk and Protective Factors for Chronic Diseases by Telephone Survey [22] indicate a correlation between lower-income and poor self-rated health status. Further, lower levels of education were associated with unhealthy behaviors such as smoking, alcohol abuse, and inadequate physical activity [18].

The consumption of UPFs is observed to be higher among individuals living in less vulnerable, higher-income areas within Brazil and in the context of this study [23]. A review of Family Budget Surveys (POF, from 1987 to 2018) shows that wealthier households demonstrate a higher tendency to buy UPFs and dine out, coupled with a decreased consumption of traditional Brazilian foods such as rice and beans [24]. This trend is mirrored in Argentine urban areas, where wealthier individuals consumed more UPF [25].

However, these findings diverge in developed countries like the USA and Canada. In these nations, UPF consumption is more prevalent among lower-income demographics and individuals facing greater vulnerability. This includes those identifying as being of Black race/skin color, those with less education, and those dealing with severe food insecurity [26,27,28].

The discrepancies between this study’s findings and those from international studies conducted in developed countries could stem from variations in economic, cultural, and food environment factors. Economically speaking, in Brazil, unlike the USA and other developed nations, UPFs are generally more expensive compared to fresh and minimally processed food [29,30]. Moreover, Brazilian eating habits mostly involve the consumption of fresh and minimally processed foods, and homecooked meals [31]. However, shifts in eating patterns among Brazilians are already evident. A review of national surveys shows that over the past decade, there has been a decrease in socioeconomic food-related inequalities, largely due to a higher increase in UPF consumption among rural, lower-income, and less-educated segments of Brazil’s population [32].

In the USA, individuals in poorer, segregated areas often face a high risk of access to UPFs, with limited availability of fresh and minimally processed options. Conversely, in Brazil, wealthier regions show a high presence of fast-food outlets and large supermarkets, offering better prices. These areas also tend to have superior infrastructure and connectivity, attributed to different economic and urban development strategies [19,33].

Different countries or cities adopt diverse policies, influencing the establishment of businesses in various regions, as well as the enforcement of food taxes, with implications for availability and pricing [34]. For instance, around the Belo Horizonte PAS in Brazil, businesses promoting immediate food consumption, largely UPFs, are prevalent in areas with a higher municipal human development index (IDHM) [35].

A further possible explanation for our study’s findings pertains to the Health Promotion Program’s potential to boost healthy eating among socioeconomically disadvantaged groups. The PAS, a critical instrument for care and health promotion purposes, aims to foster healthier communities. It is also a strategy for shrinking disparities in health service accessibility and actions encouraging healthier diets [11]. However, this program’s impact on reducing gaps in access to healthier food choices is limited. Implementing robust public policies, like improving income distribution, providing healthy food subsidies, imposing taxes on UPFs, and regulating food advertising, are essential to helping the public embrace and maintain healthier eating patterns long-term.

Although our study’s findings are significant, limitations should be considered. The first concern involves potential biases in the reported food intake due to the nature of food surveys. To minimize this, we averaged two 24HR, which were administered biannually by a trained research team and supplemented by a home measurement kit to help gauge portion sizes. Another limitation is the quality of food composition tables, which may not capture the population’s dietary variety. To mitigate this issue, we used nutritional charts of foods regularly consumed in Brazil, supplemented by food labeling datum and measured food.

One identified shortcoming was the complex task of discussing culinary preparation consumption against international studies, given that this food group remains under-researched and less commonly seen in developed countries. Studies tend to focus on the study of UPF. However, the methodological choice to group fresh and minimally processed food along with culinary ingredients into a ‘culinary preparations’ category was made to better represent Brazilian food culture and promote more precise datum interpretation. Moreover, data analysis of fresh food, culinary ingredients, and culinary preparations yielded consistent results, reinforcing the decision to retain ‘culinary preparations’ as an outcome variable.

The study has a limitation regarding its external validity since participant results from the Health Promotion Program may not reflect the general population. Still, in Brazil, PHC serves around 70% of the population [36], which can mitigate this limitation. Hence, the findings of this study can help mold public policies and actions, promoting healthy eating within the scope of Brazil’s PHC.

This study’s strengths include its large sample size and assessment of food consumption based on the NOVA classification. It applied this approach to participants from public health services, considering their health vulnerability. Moreover, the study maintained high methodological rigor, ensuring datum quality for improved internal validity.

## 5. Conclusions

The results highlight the importance of concentrating efforts on promoting and preserving healthy eating habits and emphasizing the value of homecooked meals in the most vulnerable regions. This study found that participants from less vulnerable regions require intersectoral interventions to correlate public health policies and food supply policies, aiming to curb the rise in unhealthy processed food consumption. 

## Figures and Tables

**Table 1 ijerph-21-00935-t001:** Sociodemographic characteristics of participants according to the Health Vulnerability Index.

Variable	Total (N = 3.056)		Health Vulnerability Index (HVI)						*p*-Value
			Low (n = 313; 10.2%)		Middle (n = 1733; 56.7%)		High/Very High (n = 1010; 33.0%)		
	n	Value	n	Value	n	Value	n	Value	
Sex, %									0.505
Male	372	12.2	44	14.1	211	12.2	117	11.6	
Female	2684	87.8	269	85.9	1522	87.8	893	88.4	
Age (years), median (P_25_–P_75_)	3056	58 (49–65)	313	61 (53–68) ^a^	1733	58 (50–65) ^b^	1010	56 (47–63) ^c^	<0.001 **
Marital status ^†^, %									<0.001 *
Married	1866	61.1	149	47.6	1106	63.8	611	60.6	
Separated/Divorced	258	8.4	33	10.5	129	7.4	96	9.5	
Single	446	14.6	78	24.9	221	12.7	147	14.6	
Widower	485	15.9	53	16.9	277	16.0	155	15.4	
Professional occupation, %									<0.001 *
From home	879	28.8	64	20.4	532	30.7	283	28.0	
Retired/Pensioner	1126	36.8	146	46.7	640	36.9	340	33.7	
Unemployed	62	2.0	7	2.2	32	1.8	23	2.3	
Employee	988	32.3	96	30.7	529	30.5	363	36.0	
Education (years), average (P_25_–P_75_)	3056	8 (4–11)	313	11 (5–15) ^a^	1733	8 (4–11) ^b^	1010	5 (4–10) ^c^	<0.001 **
Per capita household income ^&^, median (P_25_–P_75_)	2783	678.0 (433.3–1017.0)	289	1000.0 (633.3–2.000.0) ^a^	1562	700.0 (466.7–1.078.0) ^b^	932	600.0 (362.0–850.0) ^c^	<0.001 **
Time in PAS, median (P_25_–P_75_)	3056	16.7 (7.1–30.6)	313	11.1 (4.5–18.2) ^a^	1733	16.9 (8.4–30.5) ^b^	1010	17.9 (6.4–34.6) ^c^	<0.001 **

Note: PAS = Health Academy Program; Low HVI = 0.25–2.33; Middle HVI = 2.34–3.32; High/Very High HVI = 3.33–6.86. ^a,b,c^ Different letters indicate significant differences between categories. * Linear trend test. ** Kruskall–Wallis test with the Scheffe post hoc. ^†^ 1 missing; ^&^ 273 missing.

**Table 2 ijerph-21-00935-t002:** Health conditions and nutritional status of participants according to the Health Vulnerability Index.

Variable	Total (N = 3.056)		Health Vulnerability Index (HVI)						*p*-Value
			Low (n = 313; 10.2%)		Middle (n = 1733; 56.7%)		High/Very High (n = 1010; 33.0%)		
	n	Value	n	Value	n	Value	n	Value	
Diabetes mellitus, %	517	16.9	62	19.8	271	15.6	184	18.2	0.215
Arterial hypertension, %	1626	53.2	169	54.0	922	53.2	535	53.0	0.955
Health perception, %									<0.001 *
Very bad/Bad	17	0.6	0	0.0	7	0.4	10	1.0	
Regular	773	25.3	54	17.3	441	25.4	278	27.5	
Good very good	2265	74.1	259	82.7	1285	74.1	721	71.5	
Quality of life ^&^, %									0.002 *
Very bad/Bad	77	2.5	6	1.9	35	2.0	36	3.6	
Regular	545	17.8	49	15.6	286	16.5	210	20.8	
Good very good	2433	79.6	258	82.4	1412	81.5	763	75.6	
BMI (kg/m^2^) ^†^, median (P_25_-P_75_)	2918	27.2 (24.3–30.5)	300	26.7 (24.1–29.8) ^a^	1660	27.1 (24.2–30.5) ^b^	958	27.7 (24.6–30.8) ^c^	<0.001 **
Nutritional status ^†^, %									0.022 *
Low weight	17	0.6	3	1.0	13	0.7	1	0.1	
Eutrophy	845	27.6	98	31.3	488	28.1	259	25.6	
Overweight	1229	40.2	129	41.2	702	40.6	398	39.4	
Obesity	827	28.3	70	23.3	457	27.5	300	31.3	

Note: BMI = body mass index; Underweight = <18.5 kg/m^2^; Eutrophy = ≥18.5–24.9 kg/m^2^; Overweight = >25–29.9 kg/m^2^; Obesity= >30 kg/m^2^. Low HVI = 0.25–2.33; Middle HVI = 2.33–3.32; High/Very High HVI = 3.32–6.86. Treatment of psychiatric illnesses: anxiety, depression, nervousness, etc. ^a,b,c^ Different letters indicate significant differences between categories. * Linear trend test. ** Kruskall–Wallis test with the Scheffe post hoc. ^&^ 1 missing; ^†^ 138 missing.

**Table 3 ijerph-21-00935-t003:** Contribution of participants’ energy consumption (Kcal) according to the NOVA classification and the Health Vulnerability Index.

Variable	Total (N = 3.056)		Health Vulnerability Index (HVI)						*p*-Value *
			Low (n = 313; 10.2%)		Middle (n = 1733; 56.7%)		High/Very High (n = 1010; 33.0%)		
	n	Value	n	Value	n	Value	n	Value	
Energy consumption (kcal)									
Total	1429.7	550.5	1497.9 ^a^	617.6	1444.7 ^a^	544.5	1382.9 ^b^	535.4	0.001
Culinary preparations	873.6	398.3	887.8	461.6	874.7	391.1	867.3	389.6	0.719
Processed foods	152	130.6	159.9	132.6	157.0 ^a^	132.8	140.9 ^b^	125.7	0.004
Ultra-processed foods	404.0	302.0	450.0 ^a^	333.7	412.9 ^a^	306.6	374.6 ^b^	280.5	<0.001
% of dietary energy									
Culinary preparations	61.6	14.9	59.7 ^a^	15.7	61.1 ^b^	14.8	63.2 ^c^	14.5	<0.001
Processed foods	10.9	9.0	11.2	9.4	11.2	9.0	10.4	8.9	0.119
Ultra-processed foods	27.4	14.9	29.0 ^a^	15.6	27.8 ^b^	15.0 ^b^	26.3 ^c^	14.6	0.005

Note: Low HVI: 0.25–2.33; Middle HVI: 2.33–3.32; High/Very High HVI: 3.32–6.86. Kcal: kilocalories. ^a,b,c^ Different letters indicat significant differences between categories. * ANOVA statistical test with Scheffe correction.

**Table 4 ijerph-21-00935-t004:** Results of the multiple linear regression for the association between the Health Vulnerability Index and the percentage of energy according to the NOVA classification.

Model *	Culinary Preparations		Processed Foods		Ultra-Processed Foods	
	β (95%CI)	*p*-Value	β (95%CI)	*p*-Value	β (95%CI)	*p*-Value
Unadjusted model						
low HVI	ref.	-	ref.	-	ref.	-
middle HVI	1.3 (−0.4; 3.0)	0.150	−0.0 (−1.1; 1.0)	0.921	−1.2 (−3.0; 0.5)	0.172
high/very high HVI	3.4 (1.6; 5.3)	<0.001	−0.7 (−1.8; 0.3)	0.193	−2.7 (−4.6; −0.8)	0.005
Model 1						
low HVI	ref.	-	ref.	-	ref.	-
middle HVI	1.0 (−0.7; 2.8)	0.249	0.2 (−0.8; 1.3)	0.659	−1.2 (−3.0; 0.4)	0.157
high/very high HVI	3.0 (1.1; 5.0)	0.002	−0.2 (−1.4; 0.9)	0.689	−2.8 (−4.7; −0.9)	0.004
Model 2						
low HVI	ref.	-	ref.	-	ref.	-
middle HVI	0.9 (−0.8; 2.7)	0.318	0.3 (−0.7; 1.4)	0.510	−1.2 (−3.0; 0.5)	0.163
high/very high HVI	2.9 (0.9; 4.8)	0.003	−0.0 (−1.2; 1.1)	0.881	−2.8 (−4.7; −0.8)	0.004
Model 3						
low HVI	ref.	-	ref.	-	ref.	-
middle HVI	0.7 (−1.1; 2.5)	0.446	0.4 (−0.6; 1.5)	0.456	−1.1 (−2.9; 0.6)	0.225
high/very high HVI	2.5 (0.5; 4.5)	0.011	0.0 (−1.1; 1.2)	0.980	−2.5 (−4.5; −0.5)	0.011
Model 4						
low HVI	ref.	-	ref.	-	ref.	-
middle HVI	1.0 (−0.7; 2.8)	0.271	0.5 (−0.5; 1.6)	0.339	−1.5 (−3.3; 0.2)	0.093
high/very high HVI	2.7 (0.7; 4.7)	0.007	0.0 (−1.2; 1.2)	0.992	−2.7 (−4.7; −0.7)	0.007

Note: CI: confidence interval; Low HVI: 0.25–2.33; Average HVI: 2.34–3.32; HVI: High/Very High: 3.33–6.86. Model 1: adjusted for sex, age, and education. Model 2: adjusted by variables from Model 1 + SBP Time; Model 3: adjusted by variables from Model 2 + health perception; Model 4: adjusted by variables from Model 3 + BMI. * Multiple linear regression.

## Data Availability

The datasets analyzed during the current study are not publicly available but are available from the corresponding author on reasonable request.

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
