# Peer review of "Food Consumption Inequalities in Primary Care in a Large Metropolis"

_ijerph, 2024, doi:10.3390/ijerph21070935_

Round 1

Reviewer 1 Report

Comments and Suggestions for Authors

Dear Authors,

Congratulations to the authors for an interesting and valuable study. The study design and data analysis appear to be rigorous and appropriate.  The importance of nutrition in health and well-being is strongly recognized worldwide. This paper makes an essential contribution to the literature. 

The authors indicate the importance of implementing initiatives to encourage healthy eating, considering the vulnerability of different regions. In my opinion, What is most important is that in the most vulnerable regions, efforts should concentrate on promoting and preserving healthy eating habits and emphasizing the value of home cooking.

Generally, the results are described correctly and clearly for the reader. The selection of literature is appropriate and mostly from 2019-2024.

 I recommend shortening the titles of the tables. Information about the place and time of the study is given in the materials and methods section. 

I have a few more small comments about:

Lines 9-10 Abstract: The first two sentences lack a subject and a verb. I propose the following version:

‘According to the NOVA classification within primary care in a major Brazilian city, the study examined the link between health vulnerability and food consumption. A cross-sectional study was conducted among adults over 20 years old.’

Line 94 – You should use reference [14]

Line 101 – I doubt if the % sign is needed. If not, you should remove it;

Other minor shortcomings include:

Line 226 - remove the space after the HVI acronym

Line 243 - Sort out the order of citations, please [18,8,19].

The above-mentioned additional notes and analysis would significantly increase the value of this study. Once again, congratulations to the Authors for an interesting and valuable study.

Best Regards

Reviewer

Author Response

Thanks for the review. Attached the answers.

Reviewer 2 Report

Comments and Suggestions for Authors

Dear authors

Thank you for your important manuscript “Inequalities in food consumption in primary care in a large metropolis”. The manuscript idea is good and very important.

Methods used in collecting the data and selection participants are ideal. But if the authors take the averaged of three 24-h recalls instead of averaged two 24-h recalls in association with dietary pattern the results may be more useful.

Please correct the reference in line 94 to Willett [14].

The results were very good and discussed well.

Nutrition plays an important role in human health. However, there are many factors that affect the type and quantity of food for an individual. The most important of these factors are social level, income level, health status, and the availability of food supply chains. The economic level of countries is also one of the most important influences on the type and availability of food. It is known that wars and conflicts are among the most important factors that lead to poor food security, lack of food supply chains, and in some cases to starvation. Therefore, the topic of the current study is very important as a study that links the different influences on the availability and type of food available to different regions according to their social level, health status, and income level.

In most developing countries low levels peoples take the minimal level of nutrient. The usually used carbohydrates as the main nutrient, also they consumed little fruits and vegetables. Plant proteins are the usual source of protein in little amount.

The study concluded: The importance of implementing initiatives to encourage healthy eating, taking into account the vulnerability of different regions. This study found that participants from less vulnerable regions need intersectoral interventions to link public health policies and food supply policies, with the aim of reducing the high consumption of unhealthy processed foods. Conversely, in more vulnerable regions, efforts should focus on promoting and maintaining healthy eating habits and emphasizing the value of home cooking.

Author Response

(The authors gave the same response as above.)

Reviewer 3 Report

Comments and Suggestions for Authors

Thank you for the opportunity to review the article titled Inequalities in food consumption in primary care in a large metropolis. The authors used the NOVA classification to investigate the relationship between type of food consumed (homecooked, processed, highly processed) and a health vulnerability index allocated to the area. This is an interesting topic, particularly considering the differences between countries.

The abstract seems to be somewhat misleading by stating that the study investigated the link between health vulnerability and food consumption. This sounds like the comparisons were made based on individual health vulnerability rather than that for the area they live in. Please clarify.

The introduction provides reasonable background on the situation, explaining the importance of the investigated factors on nutrition.

The selection and matching of the PAS is somewhat unclear – perhaps you mean that units with different rather than matching HVI were selected? Why were only PAS in medium and high HVI areas selected, and how it is possible that your sample covered all categories of HVI? Additionally, the inclusion criterion of morning operation seems unnecessary and limiting.

Please clarify what the total sample is based on (L. 95) as this is not clear – total number of people invited or those who responded?

Please explain how the HVI was determined for each participant – perhaps their address?

The results section seem to confuse HVI and SVI, and the latter has not been mentioned previously. Please revise.

The comparisons in the paragraph on LL. 176-183 are somewhat unclear. Clarify whether the two comparisons are for the medium and high HVI.

L. 274-277 The relevance of this paragraph to the current study is not clear. I suggest removing it.

Terminology:

Note that the term 24-hour diet recall is preferred as it should include food and drink (and supplements, as appropriate) rather than food only. Additionally, it is not clear how the home measurement kit was used. Dietary recalls are collected retrospectively, whereas participants typically use household measures to record their intake at the time (e.g., diet record). Please check that the correct method is described.

Please explain what nutritional charts are (L. L. 303) and how you used ‘measured food’ (L. 304) to determine the type of food. Importantly, please explain how the dietary intake was classified using the NOVA criteria as the information provided in the relevant section (2.4.2.) is not sufficient to understand how each food was categorized.

Food quality can be degraded by poor storage, for example, but not socioeconomical status (L. 52-53)

L. 58 It is not clear what a healthful eating drive is – consider rephrasing and/or explaining.

BMI: the term body mass index is not capitalized; additionally, it is not measured but calculated.

Consider providing energy in SI units (kJ) rather than kcal.

In English, the word culinary is used to describe something that requires special preparation or ingredients, and seems unsuitable in this context. Consider using homecooked or homemade meals instead.

Clarity:

L. 15-16 – not clear how food recalls can be assessed on average – perhaps you mean average intake based on the diet recalls?

L. 19 Please clarify that the HVI is allocated to an area rather than a participant.

L. 33-34 note that skin color and gender cannot ‘worsen’

L. 46 Consider the wording of ethnic and racial background to enhance inclusiveness

L. 125: The abbreviation R24h has not been defined – define at first use for clarity.

L. 268 ‘consumption of food ‘accompanied by’ meals – perhaps you mean fresh food and homecooked meals here.

English language:

General: Use past tense to describe the study (e.g., L. 19, consume should be consumed)

The noun data is plural (singular: datum) and needs a plural verb (e.g., L. 121)

L. 10-11 incomplete sentence.

L. 17 should not be “Of the 3056 participants, the average…”but “The average …”

L. 45 ‘tied to’ would be best changed to ‘associated with’

L. 90 remove the comma to avoid implying that pregnant women are unable to respond to the questionnaire

L. 210 intake does not ‘stand’ (or ‘sit’), it just is. You can use ‘the intake was’

L. 261-262 revise ‘insecurit’ to insecurity

L. 315-316: something is missing in this sentence – it is not clear what the PAS influences.

Table 1. Sex is male or female. Masculine and feminine is used for grammatical gender only. I can not see all symbols for missing data defined in the footnote used in the table.

Comments on the Quality of English Language

.

Author Response

(The authors gave the same response as above.)

Reviewer 4 Report

Comments and Suggestions for Authors

MANUSCRIPT: 3083004

TITLE: Inequalities in food consumption in primary care in a large metropolis

The manuscript 3083004 “Inequalities in food consumption in primary care in a large metropolis” presents an interesting and valuable study in order to show how diet or the way in which food is consumed can affect the health of populations.

This work is well structured, well planned and the research is competently carried out and the methodology was adequate for the research.

The results were subject to appropriate statistical analysis.

The literature is well cited and most of the papers cited (75%) date back to the last five years.

Conclusions presented are in accordance with the results obtained.

Regarding the manuscript presented, I congratulate the authors for the valuable information regarding health vulnerabilities related to the type of consumption and the type of food preparation.

Regarding the manuscript presented, I only have two small questions:

1. In manuscript is used the same abbreviation (PAS) for two different designations in the manuscript. In lines 11-12, Health Fitness Program (PAS) appears, while in line 65, Health Workout Program (PAS) appears. Please correct the abbreviations because the abbreviation PAS does not seem to make sense as it does not have the same initial letters as the health programs.

2. Please include in supplementary material the questionnaires presented to individuals participating in the study.

Author Response

(The authors gave the same response as above.)
